# NHSL3 controls single and collective cell migration through two distinct mechanisms

Nikita M. Novikov [1], Jinmei Gao [2], Artem I. Fokin [1], Nathalie Rocques[1], Giovanni Chiappetta[3], Karina D. Rysenkova [1], Diego Javier Zea[2], Anna Polesskaya [1], Joelle Vinh [3], Raphael Guerois [2] & Alexis M. Gautreau [1] ✉

The molecular mechanisms underlying cell migration remain incompletely understood. Here, we show that knock-out cells for *NHSL3*, the most recently identified member of the Nance-Horan Syndrome family, are more persistent than parental cells in single cell migration, but that, in wound healing, follower cells are impaired in their ability to follow leader cells. The *NHSL3* locus encodes several isoforms. We identify the partner repertoire of each isoform using proteomics and predict direct partners and their binding sites using an AlphaFold2-based pipeline. Rescue with specific isoforms, and lack of rescue when relevant binding sites are mutated, establish that the interaction of a long isoform with MENA/VASP proteins is critical at cell-cell junctions for collective migration, while the interaction of a short one with 14-3-3θ in lamellipodia is critical for single cell migration. Taken together, these results demonstrate that NHSL3 regulates single and collective cell migration through distinct mechanisms.

Cell migration plays an essential role in the embryonic development and adulthood of animals. There are multiple modes of cell motility. In particular, cells can migrate as single cells or as groups. Migrating cells are polarised in the direction of movement. The leading edge of a migrating cell is characterised by a membrane protrusion, referred to as the lamellipodium[1]. The lamellipodium protrudes through the pushing force provided by Arp2/3-generated branched actin networks. Persistence of movement in a given direction is driven by a positive feedback loop that sustains polymerisation of branched actin, where it was previously polymerised, at the leading edge[2,3]. In collective migration, cohesiveness allows leader cells to orient follower cells via mechanotransduction through cell-cell junctions[4]. Cell-cell junctions require a dynamic actin cytoskeleton for their establishment and maintenance. A balance of pushing forces mediated by branched actin and pulling forces mediated by myosin-mediated contractility controls strength, plasticity and repair of cell-cell adhesions[5,6].

At the molecular level, the Arp2/3 complex nucleates daughter actin filaments from pre-existing filaments at the cell cortex, in lamellipodia as well as cell-cell junctions[7,8]. In both locations, its major activator is the pentameric WAVE complex[9]. The subunit WAVE that directly activates Arp2/3 is masked at resting state and becomes exposed upon signalling by the small GTPase Rac1[9]. VASP family proteins, which form tetramers clustered at the plasma membrane, increase the speed of actin polymerisation by fuelling actin monomers to the dynamic end of actin filaments[10,11]. IRSp53 is an adaptor protein that clusters VASP proteins[12]. VASP proteins operate both in lamellipodia and at cell-cell junctions[13,14]. Actin filament elongation by VASP family proteins reinforces the cytoskeletal attachment of cell-cell junctions under tension[15,16].

The Nance-Horan Syndrome gene *NHS* is mutated in children affected with facial dysmorphia, dental abnormalities, cataract and intellectual disability[17]. The NHS protein displays a WAVE homology

[1]Laboratory of Structural Biology of the Cell (BIOC), CNRS UMR7654, École Polytechnique, Institut Polytechnique de Paris, Palaiseau, France. [2]Université Paris-Saclay, CEA, CNRS, Institute for Integrative Biology of the Cell (I2BC), Gif-sur-Yvette, France. [3]Biological Mass Spectrometry and Proteomics (SMBP), ESPCI Paris, Université PSL, LPC CNRS UMR8249, Paris, France. ✉e-mail: alexis.gautreau@polytechnique.edu

domain (WHD) at its N-terminus, and the WHD defines a family of NHS-like proteins[18]. The homologous domain in WAVE proteins is critical for the assembly of the WAVE complex[19]. NHS was found to interact with WAVE complex subunits and to regulate lamellipodial protrusions[18]. The interaction of NHS family proteins with the WAVE complex was deciphered in the NHSL1 member that inhibits lamellipodial protrusions and persistence of cell migration through a two-tiered mechanism. The NHSL1 WHD domain allows the assembly of a so-called WAVE shell complex, where NHSL1 occupies the position of the WAVE subunit[20]. NHSL1 also interacts with the canonical WAVE complex, which contains WAVE, through two proline-rich motifs engaged with the SH3 domain of the ABI1 subunit[21].

Here we report the role of a functionally uncharacterised protein, KIAA1522, recently identified as a member of the NHS family protein and subsequently renamed NHSL3[22,23]. NHSL3 displays a WHD and ABI1 binding sites, similar to NHSL1. We found that NHSL3 regulates single and collective cell migration, but that none of these functions involves the WAVE complex. Using proteomics and AlphaFold2, we identify two NHSL3 isoforms and their specific effectors that control single and collective cell migration.

## Results

### NHSL3 Regulates Single and Collective Cell Migration

To investigate the potential role of NHSL3 in cell migration, we first used the untransformed epithelial cell line MCF10A derived from human breast that allows evaluating migration of cell monolayers, as well as single cells when cells are dissociated[24]. When NHSL3 was depleted using siRNAs, trajectories of single MCF10A cells became less convoluted (Fig. 1a, Supplementary Movie 1). This increase in migration persistence means that the direction of migration, chosen at random, is sustained upon NHSL3 depletion. A decrease in cell speed was associated with this increased persistence of MCF10A cells (Supplementary fig. 1a). The same effect was previously observed with a variety of perturbations of the Arp2/3 pathway in MCF10A cells[25]. We then generated NHSL3 knock-out (KO) clones in MCF10A cells using CRISPR/Cas9, as we previously described[25]. We isolated two KO clones, which expressed no NHSL3 due to premature stop codons in both alleles (Supplementary fig. 1b). The KO clones exhibited the same phenotype of increased persistence and decreased speed as NHSL3 knock-down cells (Figs. 1b, Supplementary fig. 1c, Supplementary Movie 2). We then decided to generalise these findings to another cell line. We chose the hTERT-immortalised HME1 cell line, that corresponds to untransformed human mammary epithelial cells, like the MCF10A cell line. Upon siRNA-mediated depletion of NHSL3, HME1 cells also exhibited an increase in migration persistence, that is not associated with a change in cell speed in this system (Fig. 1c, Supplementary fig. 1d, Supplementary Movie 3). Thus, we characterised the NHSL3 protein as a negative regulator of migration persistence.

We then sought to localise NHSL3 using immunofluorescence of HME1 cells, which display more pronounced lamellipodia than MCF10A cells. NHSL3 was localised in lamellipodia together with the branched actin marker cortactin (Fig. 1d)[26]. In MCF10A cells, however, NHSL3 was mostly observed at cell-cell junctions when cells touched each other (Fig. 1e). The junctional staining was lost upon NHSL3 KO. We also observed cell-cell junction staining when endogenous NHSL3 was tagged with a GFP-Flag at its C-terminus in a knock-in (KI) line derived from MCF10A cells (Supplementary figs. 2a, b). This junctional localisation prompted us to evaluate the potential role of NHSL3 in collective cell migration.

We compared NHSL3 KO clones to parental MCF10A cells in a wound healing assay. Videomicroscopy showed that all lines appeared able to close the wound (Supplementary Movie 4, Supplementary fig. 2c). We then analysed cell displacement from these phase contrast movies using Particle Image Velocimetry analysis

(PIV), which provides the field of displacement vectors and its evolution over time (Supplementary figs. 2d, Supplementary Movie 5). This analysis thereby reveals how cells respond to the mechanical stimulus by inducing coordinated migration towards the wound of the first rows of cells. Cell speed and order parameter correspond to the length and orientation of displacement vectors towards the wound. Both speed and order parameter decreased in NHSL3 KO cells compared with parental cells, especially after the first rows of cells facing the wound (Figs. 1f, Supplementary fig. 2e). The effect on cell speed was significant and more pronounced than that on the order parameter. NHSL3 is thus also a regulator of cell coordination in collective migration.

### Different isoforms control single and collective cell migration

*NHSL3* is a complex gene displaying alternative exons. We characterised the isoforms expressed in MCF10A cells using RT-PCR (Fig. 2a) and cloned them into an expression vector. NHSL3 has three possible transcription start sites, generating alternative first exons, leading to three long isoforms—because they all contain the remaining six exons—that we named i1, i2 and i3. In addition, we detected a much shorter isoform, lacking exons 5 and 6, when the third start site was used. The isoform i3 is thus either short (i3S) or long (i3L). We isolated cell lines derived from the MCF10A NHSL3 KO line that stably express either i1 (1094 amino-acids), i2 (1046 amino-acids), i3L (1035 amino-acids) or i3S (143 amino-acids). When tested in the single cell migration assay, we found, to our surprise, that only the short i3S isoform rescues the increased persistence of KO cells (Fig. 2b, c, Supplementary fig. 3a, Supplementary Movie 6, Supplementary Data 1).

Using quantitative RT-PCR (qRT-PCR), we determined that i3S mRNA is by far the least expressed isoform, amounting to only 0.1% of total NHSL3 mRNAs (Fig. 2d). The KI cell line that expresses all NHSL3 isoforms tagged at their C-terminus, allowed us to detect i3S. At the protein level, the short isoform i3S represents about 1% of total NHSL3 (Fig. 2e), since the lysate has to be diluted about 100-fold so that long isoforms generate a signal similar to the one of the short isoform in undiluted lysate.

To confirm that only the small NHSL3 isoform controls the migration persistence of single cells, we designed siRNAs that either target all long isoforms in the alternative exon 6 or that target only the small i3S isoform at its unique junction of exons 4 and 7 (Fig. 2f, g, Supplementary fig. 3b). Consistent with KO/rescue experiments, specific depletion of i3S was sufficient to generate the increased persistence phenotype previously observed when all isoforms were simultaneously inactivated by knock-down or knock-out (Fig. 2h, Supplementary Movie 7). In sharp contrast, depletion of all long isoforms was not affecting migration persistence.

The situation is markedly different for the collective migration phenotype. Indeed, the expression of each individual isoform partially rescued the phenotype of the 'slow follower cells' exhibited by the KO, but most isoforms failed to reach statistical significance. The only significant rescue was observed with the long i2 isoform (Figs. 2i, Supplementary fig. 3c). All long isoforms, but not the short i3S isoform, were localised at cell-cell junctions of MCF10A cells (Supplementary fig. 4a). In contrast, when transiently expressed into hTERT-HME1 cells, all isoforms were detected in lamellipodia (Supplementary fig. 4b). In conclusion, we detected i3S in the lamellipodium in line with its function in single cell migration and i2 at cell-cell junctions in line with its function in collective migration.

### Identification of NHSL3 Partners

We then attempted to characterise the mechanisms by which NHSL3 controls migration persistence of single cells. Since NHSL3 inhibits the persistence of MCF10A cells, as previously described for NHSL1[20,21], we combined siRNA-mediated depletion of NHSL3 and NHSL1 in MCF10A cells to test whether there would be an additive effect of NHSL3 and

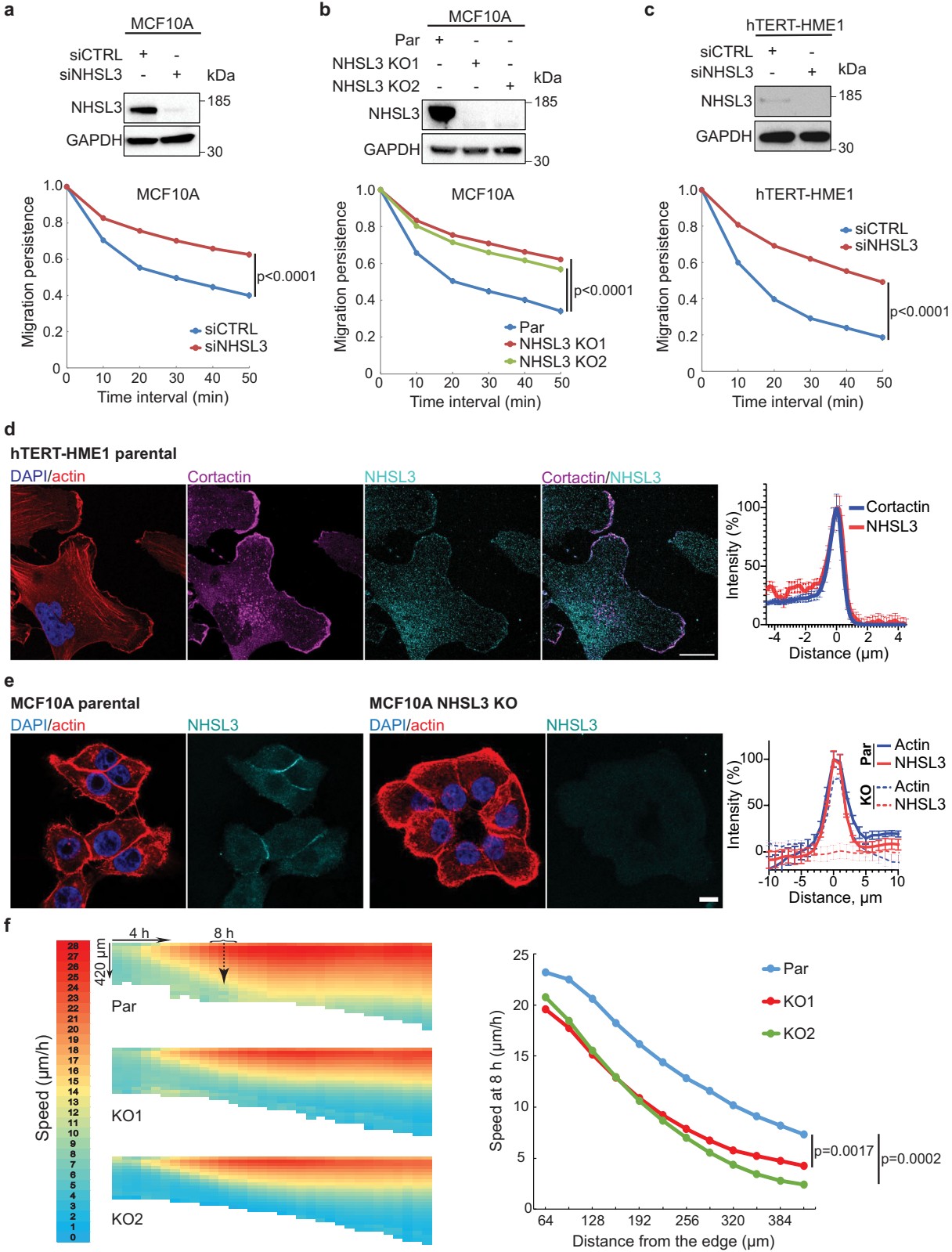

NHSL1 depletion. When both NHSL1 and NHSL3 were knocked down together, we indeed observed that migration persistence is more increased than with any single knock-down (Figs. 3a, Supplementary fig. 5, Supplementary Movie 8), suggesting that NHSL1 and NHSL3 do not control migration persistence through the same mechanism. We have previously reported that NHSL1 assembles a so-called WAVE shell complex that contains all WAVE complex subunits, except the WAVE

protein[20]. All 3 NHSL3 isoforms display homology to the WHD of NHSL1. We thus decided to clone the three N-terminal regions of NHSL3 isoforms corresponding to the NHSL1 WHD (Fig. 3b) and established MCF10A lines stably expressing NHSL3 putative WHDs or the WHD of NHSL1 as a positive control. Only NHSL1 WHD was able to associate with the WAVE complex subunits, CYFIP1, NCKAP1, ABI1 and BRK1 (Fig. 3b). Therefore, unlike NHSL1, NHSL3 does not form a WAVE

**Fig. 1 | NHSL3 regulates single and collective cell migration. a** MCF10A cells are transfected with pools of control (CTRL) or NHSL3 siRNAs and analysed by Western blots using NHSL3 and GAPDH antibodies. Cells are tracked for 6.5 h and migration persistence is extracted from trajectories of single cells. *n* = 60 cells, mean ± SEM. **b** NHSL3 KO clones or parental MCF10A cells are analysed by Western blots using NHSL3 and GAPDH antibodies. Cells are tracked for 6.5 h and migration persistence is extracted from trajectories of single cells. *n* = 60 cells. **c** hTERT-HME1 cells are transfected with pools of control (CTRL) or NHSL3 siRNAs and analysed by Western blots. Cells are tracked for 8 h and migration persistence is extracted from trajectories of single cells. *n* = 60 cells, mean ± SEM. **d** hTERT-HME1 cells are stained with DAPI (nuclear DNA), phalloidin (filamentous actin) and antibodies targeting cortactin or NHSL3. Single confocal section, scale bar: 20 μm. Overlap of cortactin and NHSL3 over multiple line scans registered to the cell edge. Data are shown as mean ± SEM, *n* = 15 line scans. **e** Parental MCF10A (Par) and NHSL3 KO cells are stained with DAPI, phalloidin and NHSL3 antibodies. Single confocal section, scale bar: 10 μm. Overlap of actin and NHSL3 over multiple line scans registered to the cell-cell junction. Data are shown as mean ± SEM, *n* = 15 line scans. **f** Migration of monolayers into the wound that is created by lifting an insert is imaged by phase contrast over time and analysed by Particle Image Velocimetry (PIV). Heat maps of the speed of collective parental MCF10A or NHSL3 KO cells display the front edge at the top and time of insert lifting at the left and its evolution over time and space (average of 12 measures, i.e. 3 biological replicates each containing 4 fields of view). Quantification of cell speed across the monolayer at 8 h. Statistical significance is calculated using custom-made R programmes for single cell migration and one-way ANOVA for collective migration. *p*-values are indicated. Three biological repeats for each experiment gave similar results. Source data are provided as a Source Data file.

shell complex. Together, these results indicated that NHSL3 controls migration persistence through a different mechanism than NHSL1.

We then sought to identify partners of NHSL3 that can account for its role in cell migration. To this end, we performed Tandem Affinity Purification (TAP) of Flag-GFP fusion proteins of the different NHSL3 isoforms expressed in stable MCF10A cell lines (Fig. 4a). We also performed TAP in the KI cell line, where all isoforms are tagged at their C-terminus with GFP-Flag (Fig. 4b). All TAPs were successful, with many partners identified for all NHSL3 isoforms using mass spectrometry (Supplementary Data 2). Logically, the short i3S isoform associates with fewer proteins, only 42, compared with the hundreds associated with long isoforms. Among partners of long isoforms, we noticed the presence of well-established regulators of actin polymerisation, such as all subunits of the WAVE complex, including WAVE2, as well as IRSp53 and MENA/VASP proteins. These actin regulators were also found to be associated with endogenously tagged NHSL3 isoforms of the KI cell line (Supplementary Data 2). To validate these partners, we performed direct GFP immunoprecipitations from the lines expressing Flag-GFP fusions proteins with each isoform. In line with mass spectrometry results, all long isoforms, but not the short i3S, were associated with these proteins involved in actin polymerisation (Fig. 4c). These results indicate these "usual suspects" might control the speed at which follower cells follow leader cells, but cannot play a role in the control of migration persistence of single cells, since they do not interact with i3S.

### NHSL3 regulates collective migration via MENA/VASP

We employed the AlphaFold2 software to predict the potential binding sites between NHSL3 and proteins of the WAVE complex, IRSp53, and MENA/VASP. As NHSL3 is predicted to be a largely intrinsically disordered protein, we used a strategy that we recently developed where proteins are cut into fragments to increase sensitivity for binding site detection[27]. The WAVE complex subunit ABI1 was predicted to bind to two sites of NHSL3 through its SH3 domain, in a manner similar to its interaction with NHSL1 (Figs. 5a, b, Supplementary Data 3)[21]. IRSp53 was also predicted to interact with NHSL3 through its SH3 domain, but to a third site. Finally, MENA/VASP proteins were predicted to interact with a fourth binding site through their EVH1 domain. Although the four predicted sites are all short proline-rich motifs, AlphaFold2 predicted binding to these different motifs in a specific manner.

We introduced into the i2 isoform the deletions corresponding to each predicted binding site and stably expressed each of these mutant forms of i2 as Flag-GFP fusion proteins in the NHSL3 KO line. GFP immunoprecipitations revealed that the binding sites of IRSp53 and MENA/VASP were correctly predicted (Fig. 5c). For ABI1, only the first predicted site appeared important for NHSL3 interaction with the WAVE complex. We then assayed collective migration of the KO line expressing mutant forms of i2 unable to interact with one or the other of these actin regulators. Only the mutant form of i2 that does no longer interact with MENA/VASP proteins (i2_Δ4) was unable to rescue the 'slow follower cells' phenotype (Fig. 5d). The mutant form of i2 that does no longer interact with the IRSp53 protein (i2_Δ3) partially

rescued the phenotype, while the mutant form of i2 that does no longer interact with the WAVE complex fully rescued it (i2_Δ1). The localisation of the mutant form of i2 that does not interact with MENA/VASP (i2_Δ4) was less pronounced in cell-cell junctions than wild-type i2 (Fig. 5e). Intensity of MENA/VASP at cell-cell junctions was also decreased in KO cells expressing this mutant form of i2 (i2_Δ4), compared with KO cells expressing wild type i2. Regarding the mutant form of i2 that does not interact with the WAVE complex (i2_Δ1), it was also less associated with cell-cell junctions than wild-type i2, and junctions were themselves less intensely stained by phalloidin in the mutant compared with the wild type, despite the full rescue the KO displayed by this mutant form (Supplementary fig. 6a). These results suggested that localisation of NHSL3 at cell-cell junctions partially depends on its interactions with both MENA/VASP and the WAVE complex. The interaction of NHSL3 with MENA/VASP proteins accounts for the recruitment of only a fraction of MENA/VASP at cell-cell junctions, but this pool appears to be critical for signal transmission to follower cells during collective migration.

### NHSL3 regulates single cell migration via 14-3-3θ

Since i3S does not interact with any of the well-established regulators of actin polymerisation, we next sought to uncover the mechanisms by which NHSL3 regulates migration persistence of single cells. i3S is likely to regulate migration persistence through the interaction with one or several of the 42 partners we identified using proteomics. We took advantage of the aforementioned in silico screening protocol by cutting the intrinsically disordered i3S into segments and isolating globular domains of partners to search for direct interactors using AlphaFold2[27]. Out of the 42 partners, six were predicted as potential direct partners of i3S C-terminal segment with a confidence score above 0.65 (Fig. 6a, Supplementary Data 3). This threshold was found as a fair trade-off between sensitivity and specificity[27]. We focused on the candidate displaying the highest confidence score, 14-3-3θ, encoded by the *YWHAQ* gene (Supplementary Data 3). Six out of seven 14-3-3 isoforms, including 14-3-3θ, also associated with long NHSL3 isoforms (Supplementary Data 2), presumably because they are phosphorylated on many residues in public databases (Uniprot & PRIDE). But 14-3-3θ was the only 14-3-3 isoform that associated with i3S.

To confirm the specific interaction of i3S with 14-3-3θ, we cloned the *YWHAQ* gene encoding 14-3-3θ and two other genes encoding 14-3-3 isoforms associated with NHSL3 long isoforms, but not with i3S, namely the *YWHAE* gene encoding 14-3-3ε and the *YWHAG* gene encoding 14-3-3γ. We transiently expressed GFP-tagged i3S and PC-tagged 14-3-3 isoforms in 293 T cells. GFP immunoprecipitates revealed that, indeed, i3S interacts with 14-3-3θ, but neither with 14-3-3ε, nor with 14-3-3γ (Fig. 6b). In the structural model generated by AlphaFold2, 14-3-3θ interacted with the unique C-terminal region of i3S that connects the amino-acids encoded by exon 4 with those encoded by exon 7. The model displayed contacts with strong physico-chemical complementary at the interface (Fig. 6c). We designed three sets of mutations, A125E, the double mutation D124A/E131A and R135D, that

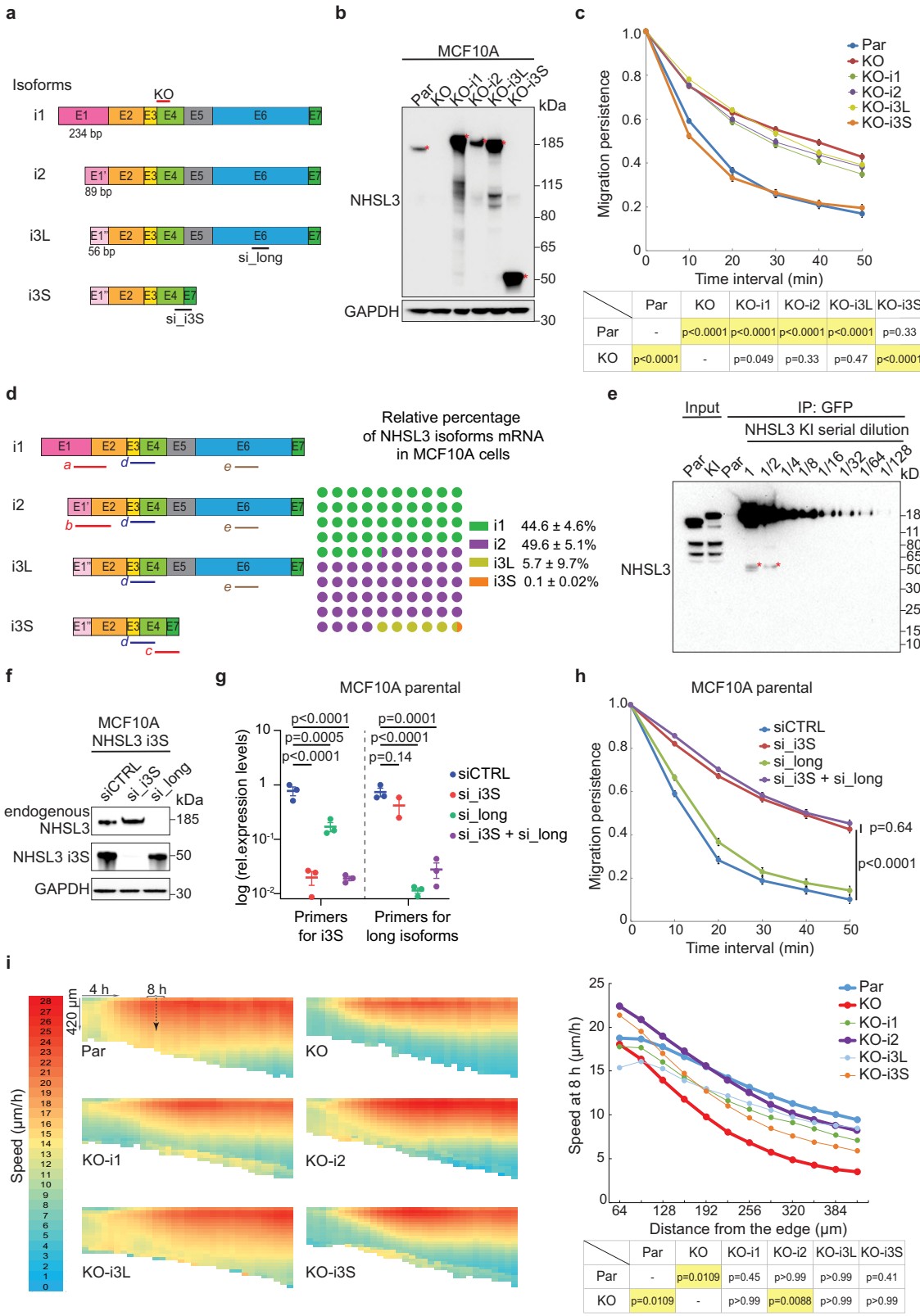

were predicted to abolish the interaction of i3S with 14-3-3θ in the structural model. When introduced into GFP-tagged i3S, all three mutations indeed abolished the interaction with PC-tagged 14-3-3θ in transiently transfected 293 T cells (Fig. 6d).

We took advantage of the availability of such mutations to examine the role of the i3S/14-3-3θ interaction in migration persistence. We focused on the R135D mutation, which is a single

substitution and that does not appear to destabilise i3S, unlike A125E (Fig. 6d). When stably expressed in NHSL3 KO cells, R135D i3S failed to interact with the endogenous 14-3-3θ expressed in MCF10A cells (Fig. 6e). Importantly, R135D i3S also failed to rescue the increased migration persistence, unlike wild type i3S (Figs. 6f, Supplementary fig. 7a, Supplementary Movie 9). If the R135D mutation renders i3S nonfunctional, because it abolishes its interaction with 14-3-3θ, then

**Fig. 2 | Different NHSL3 isoforms are involved in single and collective cell migration. a** Scheme of NHSL3 isoforms expressed in MCF10A cells. Exons 5 and 6 are omitted in a short isoform. Three transcriptional start sites account for different exons 1, labelled E1, E1′ and E1″. Locations of the gRNA used for KO and siRNAs targeting specifically short and long isoforms are indicated. **b** MCF10A NHSL3 KO cells are stably transfected with plasmids expressing Flag-GFP or Flag-GFP tagged NHSL3 isoforms and analysed by Western blots using NHSL3 and GAPDH antibodies. Red stars indicate the location of NHSL3 isoforms. **c** Cells are tracked for 6.5 h and migration persistence is extracted from trajectories of single cells. *n* = 60 cells, mean ± SEM. i3S is the only isoform that rescues the increased persistence of NHSL3 KO cells. **d** Quantification of different NHSL3 mRNAs in MCF10A cells using qRT-PCR and representation of their respective abundance. Proportions of NHSL3 isoforms are deduced from the PCR amplicons labelled a to e in the scheme, mean ± SEM. **e** Quantification of long and short isoforms at the protein level. The different isoforms are labelled at their C-terminus by knocking in GFP-Flag in exon 7

(KI). Lysates from parental MCF10A and KI cells are subjected to GFP immunoprecipitation. Lysates, immunoprecipitates and dilution thereof are subjected to NHSL3 Western blot. The short isoform is about 100-fold less abundant than long isoforms. **f** MCF10A cells stably expressing i3S are transfected with siRNAs targeting short or long isoforms and analysed by Western blots. **g** qRT-PCR of short and long isoforms in MCF10A cells transfected with siRNAs, *n* = 3, mean ± SEM. **h** Migration persistence of single MCF10A cells transfected with siRNAs. Tracking 6.5 h, *n* = 60 cells, mean ± SEM. **i** Collective migration of NHSL3 KO cells stably expressing the different NHSL3 isoforms, assessed as in Fig. 1f. i2 is the only isoform that significantly rescues the decreased speed of follower cells in NHSL3 KO cells. Statistical significance is calculated with custom-made R programmes for single cell migration or with Kruskal-Wallis test for collective migration. Significant *p*-values are highlighted in yellow in the tables. Three biological repeats of each experiment gave similar results. Source data are provided as a Source Data file.

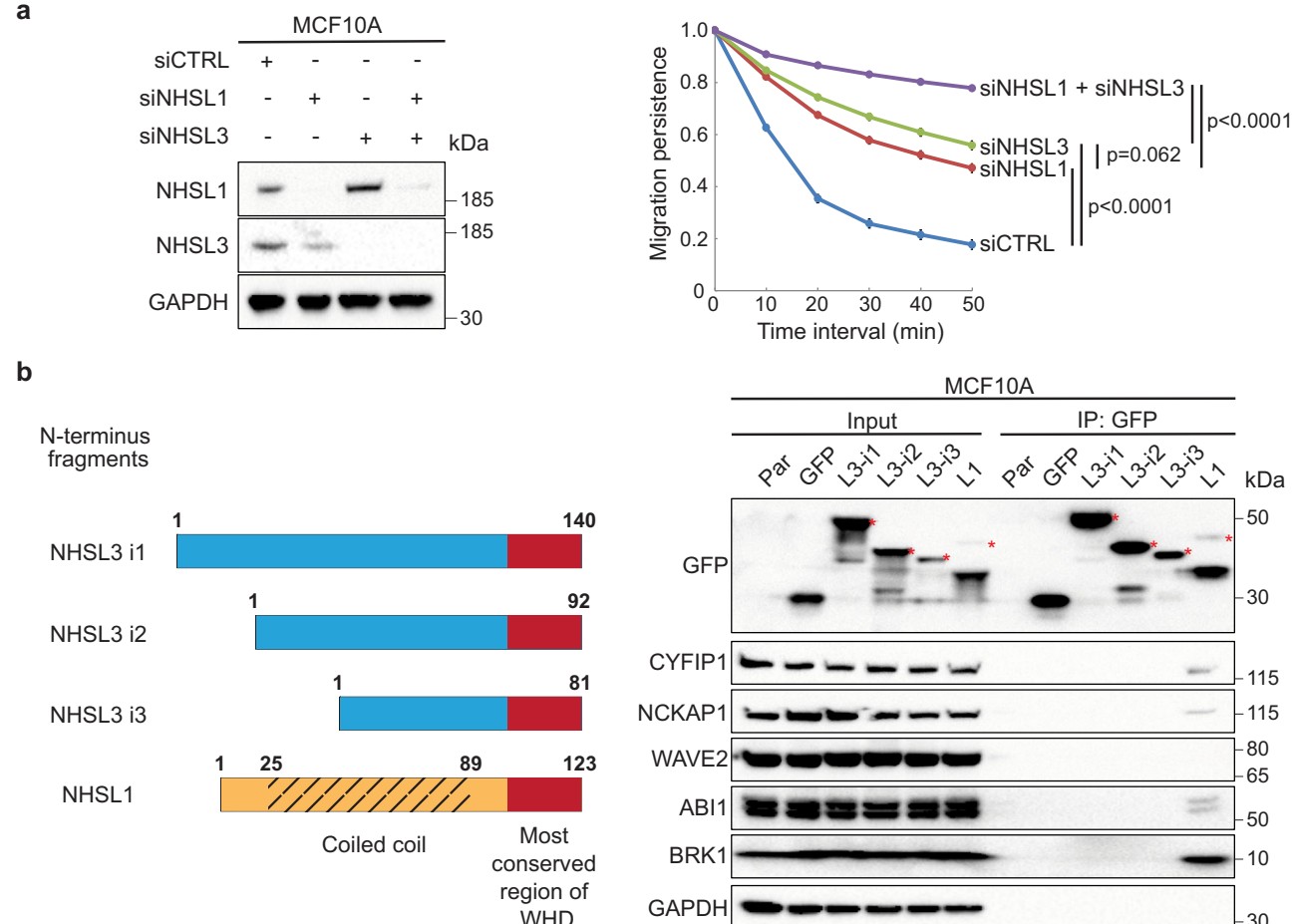

**Fig. 3 | NHSL3 regulates single cell migration through a mechanism distinct from the one of NHSL1. a** MCF10A cells are transfected with siRNA pools targeting NHSL1 and NHSL3 and analysed by Western blots. Migration persistence of single cells. Tracking 6.5 h, *n* = 60 cells, mean ± SEM. Statistical significance is calculated using custom-made R programmes for single cell migration and *p*-values are indicated. **b** The 3 N-terminal regions of NHSL3 depicted in the scheme are cloned into a plasmid expressing fusion proteins with Flag-GFP and compared with the corresponding WAVE Homology Domain (WHD) of NHSL1. Lysates of MCF10A cells

stably expressing the various fusion proteins are subjected to GFP immunoprecipitations. Lysates and immunoprecipitates are analysed by Western blots using antibodies targeting GFP, WAVE complex subunits or GAPDH as a negative control. NHSL1 with a functional WHD forms a WAVE shell complex containing all WAVE complex subunits, except WAVE2 that it replaces. None of the NHSL3 N-terminal domains forms a similar WAVE shell complex. L1 refers to NHSL1, L3-i1 to NHSL3 i1, L3-i2 to NHSL3 i2, L3-i3 to NHSL3 i3. Three biological repeats of both experiments gave similar results. Source data are provided as a Source Data file.

14-3-3θ depletion should yield the same phenotype. We thus used siRNAs to deplete 14-3-3θ from parental MCF10A, NHSL3 KO cells and KO cells stably expressing wild type or R135D i3S. As expected, 14-3-3θ depletion increased migration persistence of parental cells to the level of NHSL3 KO cells, but 14-3-3θ depletion in KO cells did not aggravate the phenotype further (Figs. 6g, Supplementary fig. 7b). Similarly, 14-3-

3θ depletion increased migration persistence of MCF10A KO cells rescued by i3S, but 14-3-3θ depletion did not aggravate the phenotype of KO cells stably expressing R135D i3S. These results indicate that 14-3-3θ is a significant partner of i3S in the control of migration persistence. We then examined the localisation of 14-3-3θ by immunofluorescence of HME1 cells. We detected a weak enrichment of 14-3-3θ

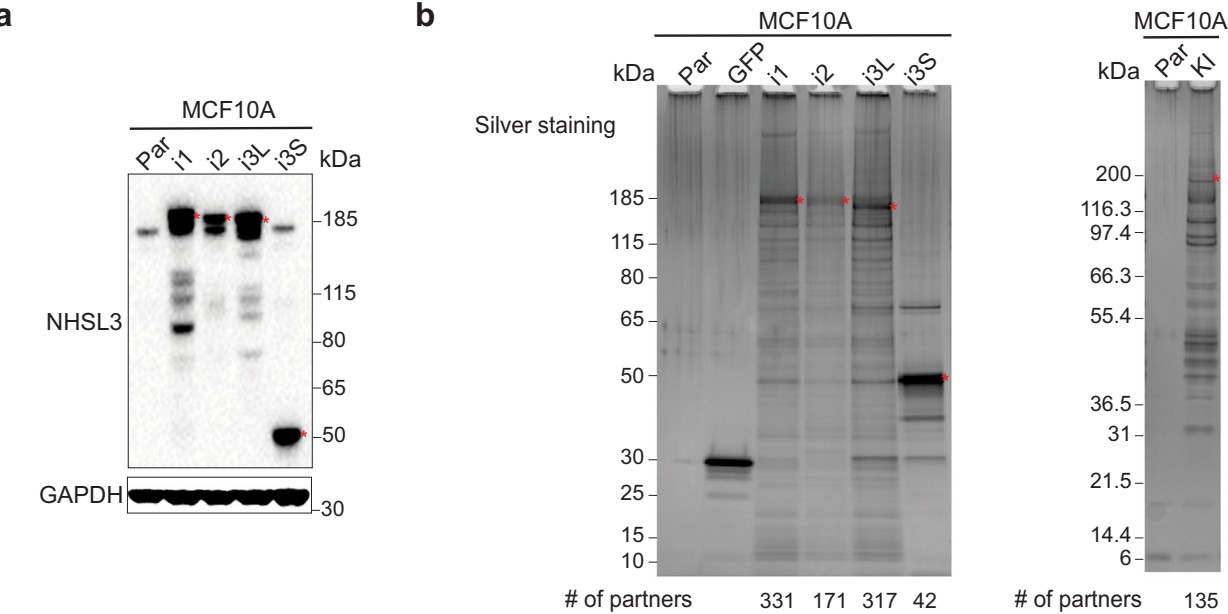

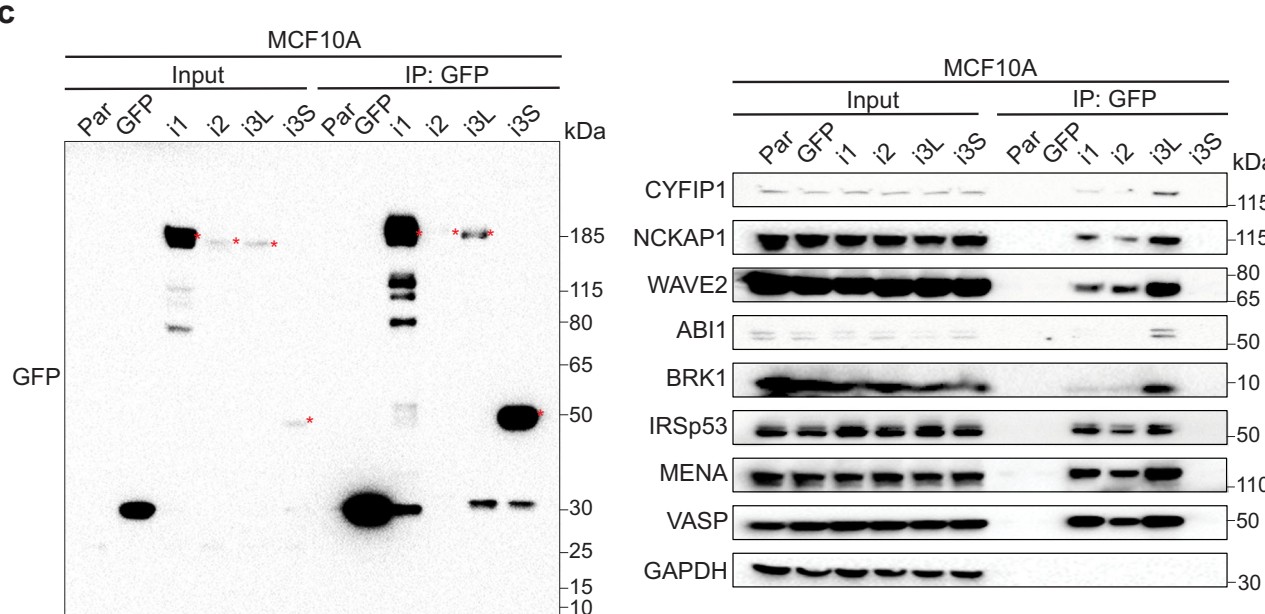

**Fig. 4 | Identification of NHSL3 partners. a** MCF10A cells are stably transfected with plasmids expressing either Flag-GFP or Flag-GFP tagged NHSL3 isoforms and analysed by Western blots. **b** FLAG-GFP tagged NHSL3 isoforms, or FLAG-GFP as a control, from stable MCF10A clones or GFP-Flag knock-in (KI) MCF10A cells are subjected to Tandem Affinity Purification (TAP). Proteins are resolved by SDS−PAGE and silver stained. The number of proteins identified by mass spectrometry is indicated below each lane. **c** GFP immunoprecipitates from MCF10A cells stably expressing NHSL3 isoforms are analysed by Western blots for the presence of partners involved in actin polymerisation, namely WAVE complex subunits, IRSp53 and MENA/VASP proteins. Red stars indicate the position of tagged isoforms of NHSL3. Three biological repeats of each experiment gave similar results.

in lamellipodia highlighted by the cortactin marker (Figs. 7a, Supplementary fig. 8a). Importantly, this lamellipodial pool of 14-3-3θ was lost upon NHSL3 depletion using siRNAs and specifically upon depletion of the short, but not the long isoforms of NHSL3 (Figs. 7b, Supplementary fig. 8b). Together, these results demonstrated that the interaction of i3S with 14-3-3θ is required for NHSL3 to recruit 14-3-3θ to lamellipodia and to control migration persistence of single cells.

## Discussion

In this work, we identified the uncharacterised NHS family member NHSL3 as a regulator of cell migration. NHS and NHSL1 had previously been reported to interact with the WAVE complex and this interaction was found to control lamellipodial protrusions and migration of single cells[18,21]. NHSL1 interacts with the whole WAVE complex through two short proline-rich motifs that interact with the ABI1 SH3 domain[21]. This mode of interaction is conserved in NHSL3, even if only one of the two predicted binding sites appears functional in NHSL3. NHSL1 also assembles a so-called WAVE shell complex through its N-terminal WHD domain, which replaces the WAVE subunit within its complex[20]. Even though the different N-terminal regions of NHSL3 isoforms exhibit homology to WAVE, like NHSL1, NHSL3 appears unable to assemble a WAVE shell complex, unlike NHSL1. The most surprising observation,

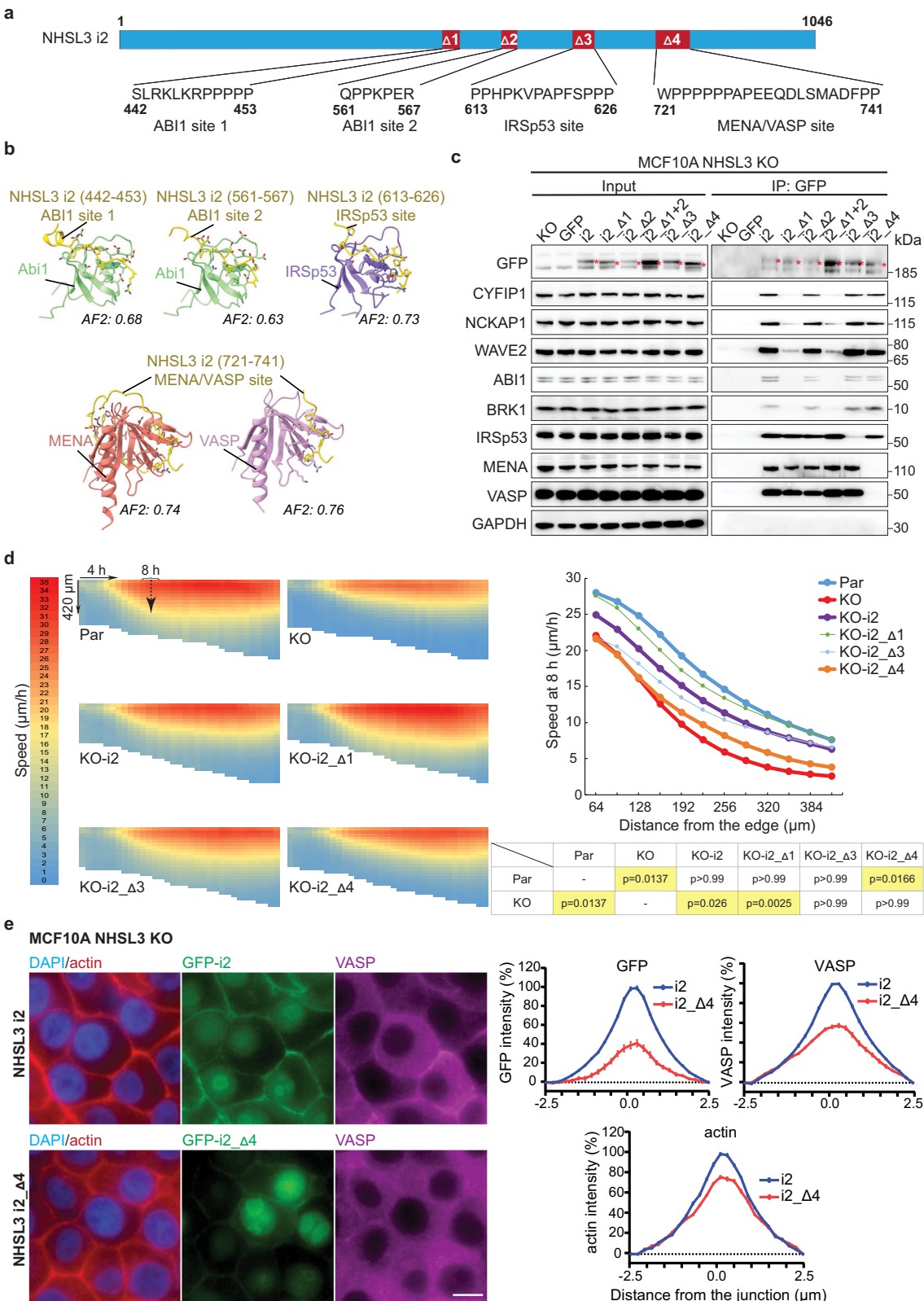

**Fig. 5 | NHSL3 i2 regulates the speed of follower cells in collective migration through its interaction with MENA/VASP family proteins. a** Predicted binding sites on NHSL3 for ABI1, IRSp53 and MENA/VASP are deleted in isoform 2, as indicated (Δ1 to Δ4). **b** AlphaFold2-generated models of interactions between NHSL3 and partners. Confidence of the interaction is indicated as the AF2 score. **c** GFP immunoprecipitates from stable rescued KO lines are analysed by Western blots. Two biological repeats with similar results. **d** Collective migration of NHSL3 KO cells stably expressing the different NHSL3 i2 mutant forms, assessed as in Fig. 1b (average of 12 measures, i.e. 3 biological replicates each containing 4 fields of view). Statistical significance is calculated with Kruskal-Wallis test and *p*-values are indicated. **e** MCF10A NHSL3 KO cells stably expressing NHSL3 i2 or NHSL3 i2_Δ4 are allowed to migrate collectively into the wound for 8 h and monolayers are stained with DAPI, phalloidin and VASP antibodies. Scale bar: 10 μm. Intensity of GFP fluorescence, VASP immunofluorescence and phalloidin is plotted as percentages from -2.5 to 2.5 μm across the cell-cell junction (mean ± SEM of 45 measures, i.e. 3 biological replicates each containing 15 line scans). The three biological repeats of all displayed experiments gave similar results. Source data are provided as a Source Data file.

we believe, is that even though NHSL3 interacts with the WAVE complex, this interaction does not appear to play any role in the dual regulation of single and collective cell migration that NHSL3 exerts.

NHSL3 regulates a specific aspect of collective migration. During wound healing, NHSL3 inactivation induces a migration defect in follower cells, but not in leader cells, since leader cells are moving forward at the same speed in NHSL3 KO and parental cells (Supplementary fig. 2c). This phenotype suggests that NHSL3 is not involved in wound sensing and initial migration induction, but rather that NHSL3 might be involved in transmitting the mechanical signal from the wound edge to cells further back in the monolayer, to induce coordinated migration. The localisation of NHSL3 in cell-cell junctions would be consistent with this interpretation (Fig. 8). NHS and NHSL1 also localise to cell-cell junctions[18,28], but these two family members were not reported to regulate collective cell migration. The localisation of NHSL3 at cell-cell junctions is specific to long isoforms, which exhibit the proline-rich motifs that interact with the WAVE complex and MENA/VASP proteins. The WAVE complex multiplies actin filaments through Arp2/3-mediated nucleation of branched actin, whereas MENA/VASP proteins elongate actin filaments from their barbed end[29,30]. VASP family proteins are thought to elongate filaments from the Arp2/3-generated branched actin networks in lamellipodia[31]. Lamellipodin is another example of a protein that coordinates these two molecular machines for efficient cell migration and invasion of cancer cells[32,33]. However, functionally, the interaction of NHSL3 with MENA/VASP proteins, but not with the WAVE complex, is required to rescue the phenotype of follower cells exhibited by KO cells. Therefore, NHSL3 regulates collective migration through its interaction with MENA/VASP proteins.

AlphaFold2-mediated prediction of protein-protein interactions[34,35] played a critical role in this study, not only to predict binding motifs of long isoforms, because these binding sites to SH3 domains and EVH1 were already described[22,36], but especially to screen partners identified by proteomics. Partners identified by proteomics are too numerous to screen them all functionally. AlphaFold2 permits to distinguish potential direct partners among those and predicts the associated binding site. The predictive pipeline we have developed, where protein partners are cut into domains before being tested, involving the fragmentation of the protein partners in the disordered regions, significantly increases the success rate of protein-protein interaction detection[27]. This approach allowed us to specifically focus on 14-3-3θ, because 14-3-3θ displayed the best confidence score. 14-3-3θ is not a protein obviously involved in cell migration, even if other 14-3-3 family members were previously shown to regulate cell migration[37].

14-3-3θ fulfilled all the expected criteria for a significant partner of NHSL3 controlling single cell migration. siRNA-mediated depletion of 14-3-3θ phenocopied NHSL3 depletion and mutation of 14-3-3θ binding site on NHSL3 prevented i3S from rescuing the phenotype of KO cells. 14-3-3 family proteins primarily function as dimers that recognise phosphorylated residues, even if this is not an absolute requisite[38–40]. 14-3-3 family proteins can heterodimerize, yet we only found 14-3-3θ associated with i3S by proteomics and this specificity among 14-3-3 proteins was confirmed using co-immunoprecipitations with other expressed 14-3-3 proteins. Since potentially phosphorylated residues were not included in the structural predictions of i3S interaction with its partners, it is not sure that the 14-3-3θ interaction with i3S requires a phosphorylated residue in NHSL3. The structural model that Alpha-Fold2 predicted involved the unique junction of exon 4 with exon 7 in i3S and the phospho-binding groove on 14-3-3θ. This model was experimentally tested using several mutations that indeed abrogated the interaction. Therefore, even if the single 14-3-3θ subunit engaged with i3S does not require a phosphorylated residue, the other 14-3-3θ subunit of the dimer would still be available to engage with a phosresidue of NHSL3 or one of its partners. The interaction of i3S with 14-3-3θ was clearly required to recruit a pool of 14-3-3θ to lamellipodia.

This pool appeared minor in quantity, yet, functionally, we have shown that it was critical to control single cell migration.

We have previously extensively characterised the molecular mechanisms of migration persistence in single cell migration in the MCF10A cell system and found that Arp2/3 activation by the WAVE complex and the branched actin it generates at the lamellipodial edge were essential[24,25,41,42]. Here, we uncovered a regulator of migration persistence, NHSL3, that can bind to the WAVE complex, but employs other molecular machines to control migration. NHSL3 regulates both single and collective cell migration through distinct isoforms. We were able to identify 14-3-3θ and MENA/VASP proteins as essential partners of NHSL3 for the control of single and collective cell migration, respectively. Our work highlights the power of AlphaFold2-mediated structural predictions in deciphering the molecular mechanisms of a protein of interest and the importance of NHS family proteins in regulating cell migration through various mechanisms.

## Methods

### Cell culture

MCF10A cells were grown in DMEM/F12 medium supplemented with 5% horse serum, 20 ng/mL epidermal growth factor, 10 μg/mL insulin, 100 ng/mL cholera toxin, 500 ng/mL hydrocortisone and 100 U/mL penicillin. hTERT-HME1 cells and HEK293T cells were grown in RPMI1640 or DMEM medium, respectively, with 10% FBS and 100 U/mL penicillin/streptomycin. hTERT-HME1 (ME16C) and HEK 293 T (CRL-3216) cells were from ATCC. MCF10A cell line was from the collection of breast cell lines organised by Thierry Dubois (Institut Curie, Paris). Media and supplements were from Life Technologies and Sigma. Cells were incubated at 37 °C in 5% CO2. Cells were passaged every 3 days.

### Plasmids, transfection and isolation of stable cell lines

Flag-GFP tagged proteins were expressed from a home-made vector, MXS AAVS1L SA2A Puro bGHpA EF1Flag GFP Blue SV40pA AAVS1R that was previously described (Molinie et al., 2019). Full-length NHSL3 isoforms (i1 NP_065939, i2 NP_001356482, i3L NP_001185901 and i3S NP_001185902), as well as N-termini of NHSL3 and NHSL1 (i1_1-140 aa, i2_1-92 aa, i3_1-81 aa, NHSL1_1-123 aa) and mutants of NHSL3 i3S (i3S_A125E, i3S_D124A/E131A, i3S_R135D) and NHSL3 i2 (i2_Δ1 lacking 442-453 aa, i2_Δ2 lacking 561-567 aa, i2_Δ1 + 2 lacking 442-453 aa and Δ561-567 aa, i2_Δ3 lacking 613-626 aa, i2_Δ4 lacking 721-741 aa) were amplified using primers detailed in Supplementary Data 4 and inserted into MXS AAVS1 vector in place of the Blue cassette using Fse1 and Asc1 restriction sites.

Transfections were performed using Lipofectamine 3000 (Invitrogen). To obtain stable cell lines in MCF10A, MXS AAVS1 vectors were co-transfected with two TALEN constructs (Addgene #59025 and 59026) inducing a double-strand break at the AAVS1 locus[43]. Cells were selected with 0.5 μg/mL puromycin (Invivogen). Single clones were picked using cloning rings, expanded and analysed by Western blot.

### Antibodies

The antibodies used were: anti-NHSL3 (Sigma-Aldrich, HPA064839, 1:1000); anti-NHSL1 (Sigma-Aldrich, HPA029967, 1:1000); anti-GFP (Roche, 11814460001, 1:1000); anti-NCKAP1 (Bethyl Laboratories, A305-178A, 1:2000); anti-MENA (Sigma-Aldrich, HPA028696, 1:1000); anti-VASP (Sigma-Aldrich, HPA005724, 1:1000); anti-IRSp53 (Proteintech, 11087-2-AP, 1:1000); anti-14-3-3θ (Bethyl Laboratories, A303-146A, 1:1000); anti-GAPDH (Thermo Fisher Scientific, AM4300, 1:4000), anti-α-tubulin (Sigma T9026, 1:2000) and anti-cortactin (Sigma 05-180-I-100UL, 1:200). Home-made CYFIP1, ABI1, WAVE2 antibodies and BRK1 antibody were described previously[44,45].

### Western blots

Cells were lysed in RIPA buffer (HEPES 50 mM, EDTA 10 mM, SDS 0.1%, NP-40 1%, DOC 0.5%, NaCl 15 mM, pH7.4) supplemented with protease

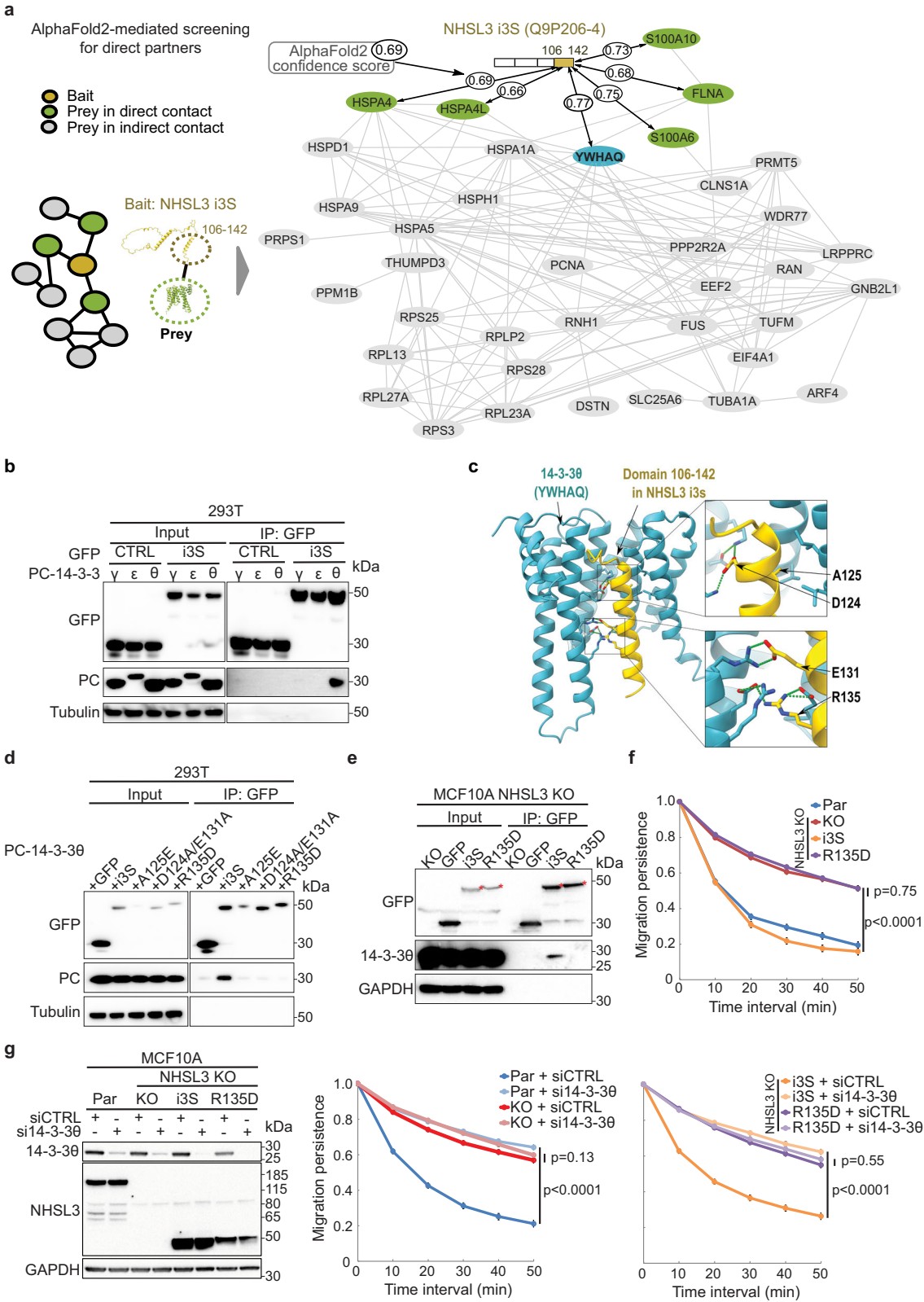

inhibitors (Roche), the lysates were clarified by centrifugation at 14,000 rpm for 15 min and subjected to SDS−PAGE using NuPAGE 4-12% Bis-Tris or 3–8% Tris-Acetate gels (Life Technologies). Pieces of nitrocellulose membranes were cut and incubated with different primary antibodies, HRP conjugated secondary antibodies (Sigma) and developed with SuperSignal™ West Femto Substrate (Thermo Fisher

Scientific) and ChemiDoc imaging system (Bio-Rad). Uncropped images are provided in the source data file.

### qRT-PCR

For qRT-PCR, total RNA from MCF10A parental cells or from MCF10A parental transfected with siCTRL, si_i3S, si_long or si_i3S and si_long

**Fig. 6 | The short isoform of NHSL3 regulates migration persistence of single cells through its interaction with 14-3-3θ. a** Prediction of direct partners of NHSL3 i3S using an AlphaFold2-based pipeline that decomposes tested proteins into evolutionary conserved domains. Partners bound to the unique C-terminal domain of i3S are shown. *YWHAQ* encoding 14-3-3θ displays the highest confidence score. Grey connections indicate protein-protein interaction reported in Biogrid. **b** 293 T cells are transiently co-transfected with plasmids expressing GFP-tagged NHSL3 i3S and PC-tagged 14-3-3 family proteins. Lysates and GFP immunoprecipitates are analysed by Western blots. **c** AlphaFold2-generated model of the interaction between NHSL3 i3S and 14-3-3θ. i3S residues that are critical for the interaction are displayed in full. **d** 293 T cells are transiently co-transfected with plasmids expressing the PC-tagged 14-3-3θ and the GFP-tagged i3S containing the indicated mutations. Lysates and GFP immunoprecipitates are analysed by Western

blots. **e** Lysates from NHSL3 KO and NHSL3 KO stably expressing GFP or GFP-tagged NHSL3 i3S or the R135D derivative thereof are subjected to GFP immunoprecipitation and analysed by Western blots. **f** Migration persistence extracted from single cell trajectories of the same cell lines. Tracking 6.5 h, *n* = 60 cells. **g** Parental MCF10A, NHSL3 KO and KO cells stably expressing i3S or the R135D derivative are transfected with control or 14-3-3θ targeting siRNAs. Migration persistence is extracted from single cell trajectories and lysates from siRNA-transfected cells are analysed by Western blots. Tracking 6.5 h, *n* = 60 cells. Data are shown as mean ± SEM. Statistical significance is calculated with custom-made R programmes for single cell migration and *p*-values are indicated. Three biological repeats of each experiment gave similar results. Source data are provided as a Source Data file.

was extracted by NucleoSpin RNA Plus Kit (Macherey-Nagel). To generate cDNA, 1.2 μg of total RNA and SuperScript III first strand synthesis for RT-PCR kit (Invitrogen) was used. PCR was carried out using iTaq Universal SYBR Green Supermix (Bio-Rad) on a CFX96 (Bio-Rad). The thermal cycling programme for template denaturation, primer annealing and primer extension was 95 °C for 10 sec, 60 °C for 30 sec for 35 cycles. To calculate the ratio of each splice isoform i1, i2, i3S, i3L of NHSL3 we used the following formula: $2^{-\Delta Ct}$, where $\Delta Ct$ = (Ct particular isoform mRNA − Ct total NHSL3 mRNA in the same sample)[46]. Nucleotide sequences of the primers used are provided in Supplementary Data 4.

### Knock-down, knock-out and knock-in
MCF10A and hTERT-HME1 knockdown cells were obtained by transfecting 10 or 20 nM siRNAs with Lipofectamine RNAiMAX (Invitrogen). siRNAs were Dharmacon ON-TARGET SMART Pools (L-010728-02-0005 for KIAA1522/NHSL3, L-032698-00-0010 for NHSL1, L-012329-00-0005 for YWHAQ/14-3-3θ) or Dharmacon ON-TARGET single siRNAs (J-010640-05-0002 and J-010640-08-0002 for NCKAP1).

Knock-down of specific NHSL3 isoforms was obtained by transfecting 10 nM of the following siRNAs (Sigma) with Lipofectamine RNAiMAX (Invitrogen):

siCTRL: 5′-AAUUCUCCGAACGUGUCACGUUU-3′;
si_long: 5′-GGAGUGUGUCCCUGCGUAAdTdT-3′ for long isoforms of NHSL3;
si_i3S: 5′-CAGCGCAAAGACUCACAGAAAdTdT-3′ for NHSL3 i3S.
Cells were analysed after 3 days.

MCF10A NHSL3 knockout cell lines were generated with CRISPR/Cas9 system. The following gRNAs were used: NHSL3 5′- CAGUCA-GACCACAUCCUACG-3′, non-targeting 5′- AAAUGUGAGAUCAGA-GUAAU-3′. Cells were transfected with the gRNA:tracrRNA duplex and purified Cas9 protein using Lipofectamine CRISPRMAX™ (all reagents from ThermoFischer Scientific). After 2 days, cells were diluted at 0.8 cells/well in 96-well plates. Single clones were expanded and analysed by Western blot. The alleles of KO clones were characterised by sequencing.

MCF10A NHSL3 knock-in cell lines were generated with CRISPR/Cas9 system. The following gRNA that targets genomic DNA, in exon 7, at the level of the stop codon of *NHSL3* was used: NHSL3 C-terminus 5′-CUGACCACCAGGCACCUCAC-3′. Corresponding oligonucleotides were annealed and cloned in the BsaI restriction site of pRG2(-GG) plasmid. The donor plasmid was constructed as follows. The NHSL3 homology arm left (HL) and homology arm right (HR) flanking Cas9-targeted site were synthesised by Integrated DNA Technologies. The GFP-Flag was amplified from the custom-made plasmid MXS GFP using primers containing the Flag sequence. The T2A-Puro was amplified from the custom-made plasmid MXS Puro using primers containing the T2A sequence. The donor cassette was constructed by assembling HL, GFP-Flag, T2A-Puro, and HR by MXS chaining[47]. MCF10A cells were transfected with the hSpCas9 CMV noSelectionMarker bGHpA and gRNA containing pRG2(-GG) plasmids together with the NHSL3 HL-

GFP-Flag-T2A-Puro-HR donor plasmid described above. Transfection was performed by electroporation at 240 V and 950 μF using ECM 630 system (BTX). Cells were selected with 0.5 μg/mL of puromycin (InvivoGen). Single clones were picked with cloning rings, expanded and analysed by Western blot and Sanger sequencing.

### Immunofluorescence
Cells were seeded on glass coverslips coated with 20 μg/mL bovine fibronectin (Sigma) for 1 h at 37 °C. Cells were fixed in 3.2% PFA on PBS, permeabilised with 0.5 % Triton X-100, blocked in 2% BSA and incubated with antibodies (1-5 μg/mL for the primary, 5 μg/mL for the secondary) or 1:3000 diluted SiR-actin (Tebu-bio). Nuclei were counterstained with DAPI (Live Technologies). Images were acquired by Axio Observer microscope (Zeiss) or when indicated, on the SP8ST-WS confocal microscope equipped with a HC PL APO 63x/1.40 oil immersion objective, a white light laser, HyD and PMT detectors. Image analysis was performed with ImageJ FIJI software. To compare fluorescence intensity in different conditions in an experiment, images were acquired with the same exposure time and microscope settings.

### Immunoprecipitation and tandem affinity purification
MCF10A cells stably expressing NHSL3 GFP-Flag KI or Flag-GFP NHSL3 isoforms were lysed with XB-NP40 buffer (50 mM HEPES, 50 mM KCl, 1% NP-40, 10 mM EDTA, pH 7.7) supplemented with protease inhibitors at 4 °C for 30 min. The lysates were clarified by centrifugation at 13,000 rpm for 15 min.

For TAP, clarified cell extracts were incubated with FLAG-M2 beads (Sigma) at 4 °C for 4 h. FLAG-M2 beads were washed with XB-NP40 buffer and eluted with 0.5 mg/ml FLAG peptide (Sigma) in XB (50 mM HEPES, 50 mM KCl, 10 mM EDTA, pH 7.7) overnight at 4 °C. FLAG eluates were collected and incubated with GFP-trap beads (Chromotek) at 4 °C for 1 h. The GFP-trap beads were washed with XB-NP40 buffer. 20% of the beads were subjected to SDS–PAGE for silver staining (SilverQuest Silver Staining Kit, Thermo Fisher Scientific) and 80% of the beads were analysed by mass spectrometry.

For GFP immunoprecipitation clarified cell extracts were incubated with GFP-trap beads (Chromotek) at 4 °C for 1 h. The GFP-trap beads were washed with XB-NP40 buffer. Beads were subjected to SDS–PAGE for Western blot.

### Mass spectrometry
The resins containing the immunoprecipitated samples were loaded onto a 10 kDa cutoff centrifugal filters (Microcon, Millipore-Merck) and washed with 500 μL of buffer (50 mM, pH 8.0, AMBIC). Disulphide reduction was performed by adding to the centrifugal filters 200 μL of a solution containing 10 mM dithiothreitol in AMBIC for 2 h at 37 °C. Thiol alkylation was performed by adding to the previous samples 20 μL of a solution containing 500 mM iodoacetamide in AMBIC for 30 minutes at room temperature. Reagents were removed by filtration and sample washed three times with 500 μL of AMBIC. Proteins were digested with 0.5 μg of trypsin/Lys-C (Promega) in 100 μL of AMBIC

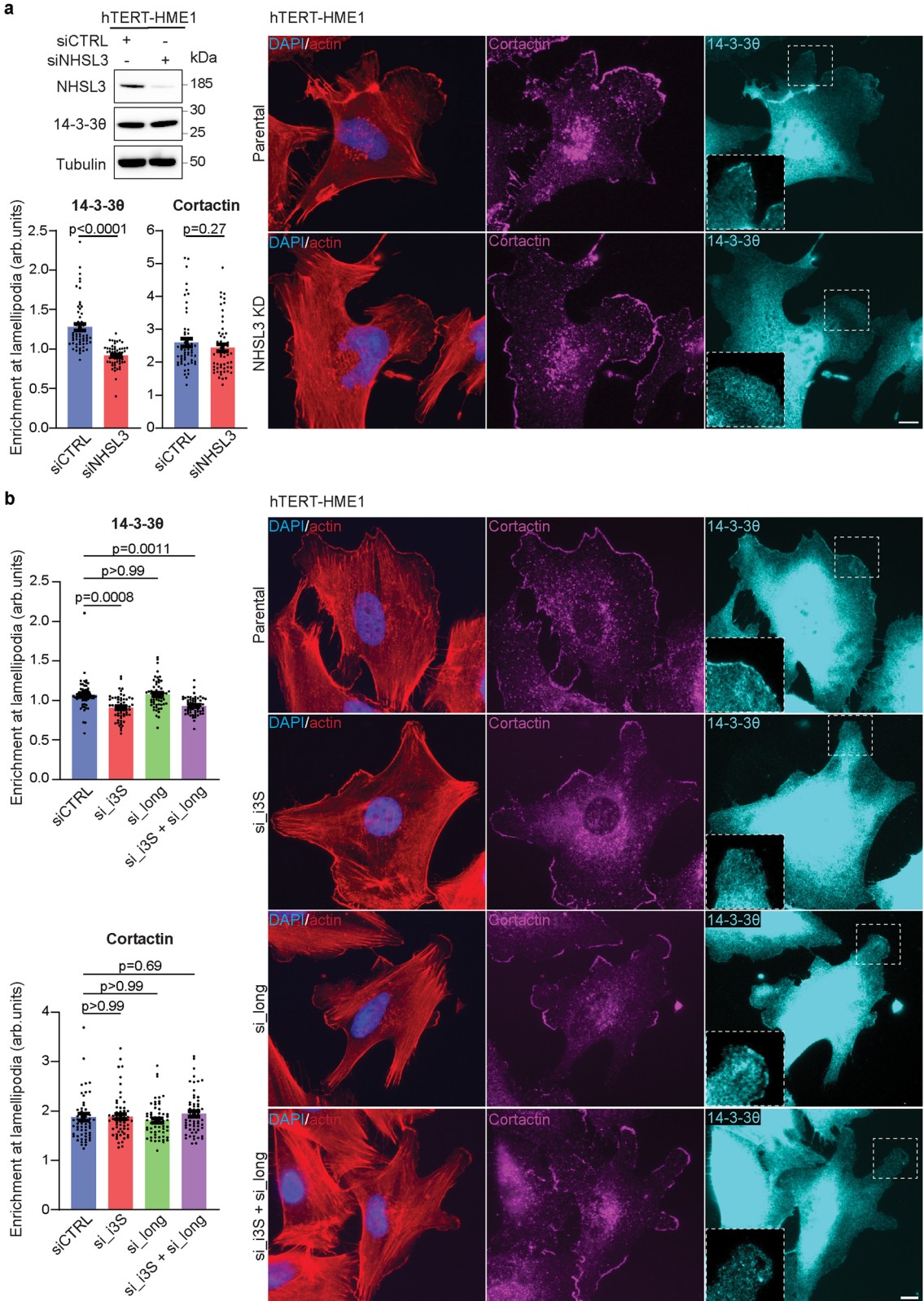

**Fig. 7 | The short isoform of NHSL3 recruits 14-3-3θ to lamellipodia.** hTERT-HME1 cells are transfected with siRNAs targeting all NHSL3 forms (**a**) or specifically long and short isoforms (**b**). Cells are subjected to immunofluorescence with DAPI, phalloidin and antibodies recognising 14-3-3θ or cortactin. Enrichment of 14-3-3θ and cortactin at the lamellipodial edge is measured and plotted as the fold increase compared with signal intensity of a similar region of interest beneath the plasma membrane, mean ± SEM. Scale bars: 10 μm. n = 57 cells for (**a**) and n = 60 cells for (**b**). Statistical significance is calculated with two-sided Mann-Whitney test (**a**) or Kruskal-Wallis test (**b**) and p-values are indicated. Three biological repeats of both experiments gave similar results. Source data are provided as a Source Data file.

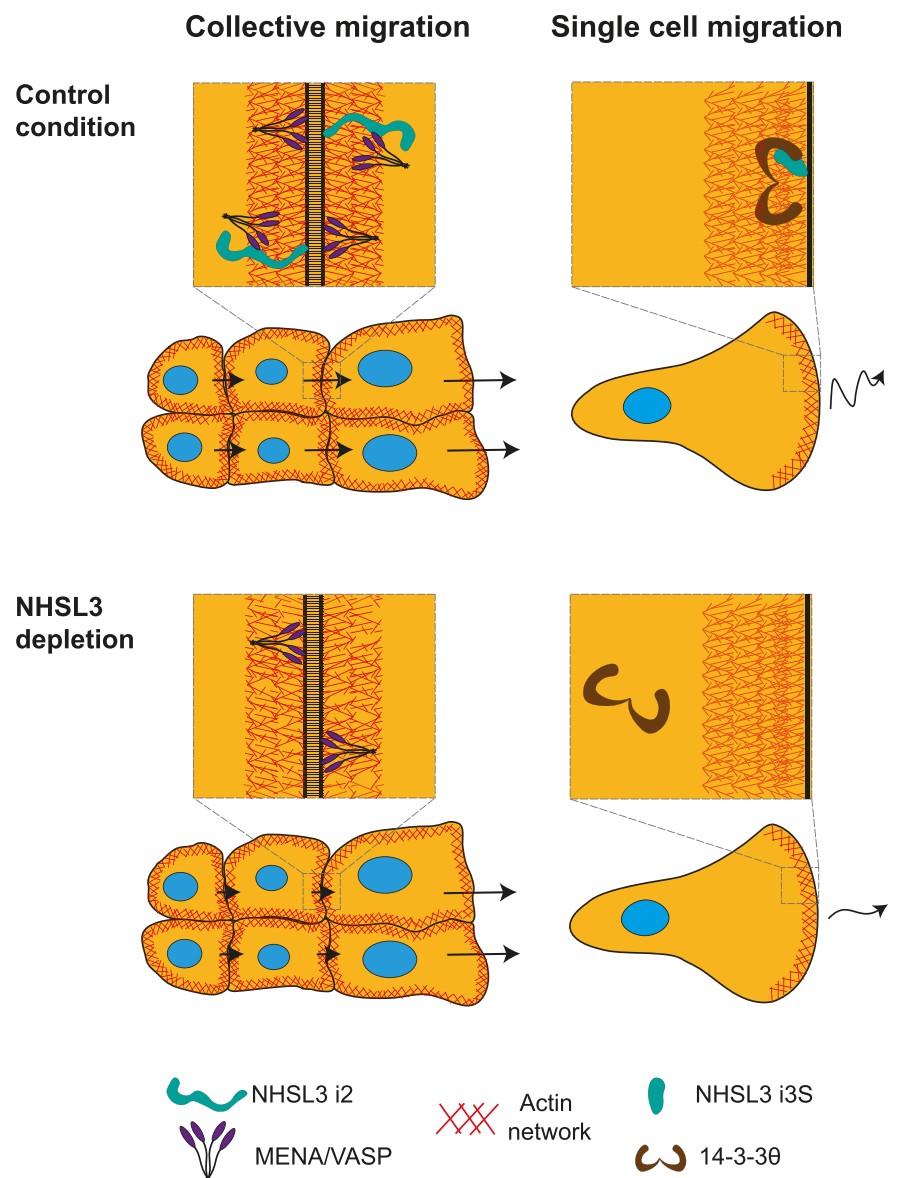

**Fig. 8 | Model.** NHSL3 regulates single and collective cell migration through two distinct mechanisms. The long isoform i2 regulates speed of follower cells in collective migration by recruiting a pool of MENA/VASP proteins to cell-cell contacts in migrating monolayers. The short isoform i3S regulates migration persistence of single cells by recruiting 14-3-3θ to the lamellipodium.

overnight at 37 °C. The resulting peptide mixture was filtered and acidified with trifluoro acetic acid at a final concentration of 0.1%. Technical triplicates were systematically analysed.

For each fraction, 6 µL of sample was concentrated on a C18 cartridge (Dionex Acclaim PepMap100, 5 µm, 300 µm i.d. x 5 mm) and eluted on a capillary reverse-phase column (nanoEaze M/Z Peptide CSH C18, 1.8 µm, 75 µm i.d. x 25 cm) at 220 nL/min, with a gradient of 2% to 38% of buffer B in 60 min (buffer A: 0.1% aq. Formic Acid/Acetonitrile 98:2 (v/v); buffer B: 0.1% aq. Formic Acid/Acetonitrile 10:90 (v/v)), coupled with a quadrupole-Orbitrap mass spectrometer (Q Exactive HF, ThermoFisher Scientific) using a Top 20 data-dependent acquisition MS experiment: 1 survey MS scan (400-2,000 m/z; resolution 70,000) followed by 20 MS/MS scans on the 20 most intense precursors (dynamic exclusion of 30 s, resolution 17,500).

Protein identification was performed with MaxQuant search engine v.1.5.3.30 against the human Swiss-Prot database (updated in 05/2022), with the following parameters: methionine oxidation, asparagine/glutamine deamidation as variable modifications, cysteine carbamidomethylation as fixed modification, first search error tolerance 20 ppm, main error tolerance 6 ppm, MS/MS error tolerance 20 ppm, FDR 1%. Quantification was performed in label-free LFQ normalisation mode[48] using at least 2 razor or unique peptide per protein. Quantities were estimated using LFQ intensities. Proteins detected in at least 2 of 3 replicates, and not detected in any of the control samples, were considered specific NHSL3 partners.

**Structural modelling**

The structural modelling of protein complexes was performed using AlphaFold2[34] with a fragmentation strategy that increases the sensitivity of detection[27]. Sequences of NHSL3 i2, i3S, ABI1, IRSp53, MENA, VASP and of the proteins detected as partners of i3S using proteomics were retrieved from the UniProt database and were submitted to three iterations of MMseqs2[49] against the uniref30_2202 database. The resulting multiple alignments of orthologs were filtered using hhfilter[50] using the following parameters ('id'=100, 'qid'=25, 'cov'=50) and the taxonomy assigned to every sequence, keeping only one sequence per

species. Full-length sequences in alignments were then retrieved and these sequences were realigned using MAFFT[51] with the default FFT-NS-2 protocol.

The NHSL3 i2 isoform was fragmented into 15 overlapping fragments, each approximately 100 amino acids long which were predicted against the different domains detected in the four partners ABI1, IRSp53, MENA, and VASP (Supplementary Data 3). NHSL3 i3S was fragmented into four domains (1–41, 32–95, 86–115 and 106–142) used as baits and predicted against 42 prey partners either fragmented into domains or considered as full-length proteins (Supplementary Data 3). To build the so-called mixed co-alignments, sequences in the alignment of individual partners were paired according to their assigned species and left unpaired in case no common species were found[52]. The concatenated multiple sequence alignments of tested partners were used as input of 1 run of the AlphaFold2 algorithm with 3 recycles each and 5 generated models[34] using ColabFold v1.5.2[52] with the Multimer v2.3 model parameters[35] on NVidia A100 GPUs.

Four scores, the pLDDT, the pTMscore, the ipTMscore and the model confidence score (weighted combination of pTM and ipTM scores with a 20:80 ratio), were provided by AlphaFold2 to rate the quality of models. The solutions obtained with NHSL3 i2 fragments were initially ranked using the average ipTM score calculated over the five models generated by AlphaFold2. The scores obtained for all the generated models are reported Supplementary Data 3. After identifying fragments containing the most probable binding site, final models were generated using AlphaFold2 (5–25 models), restricting the delimitations to the regions in contact and selecting the model with the highest confidence score for minimisation for release in the ModelArchive database.

## Migration and videomicroscopy

All cell migration assays were performed in μ-Slide eight-well dishes (#80826, Ibidi). For single cell migration assays, cells were seeded on dishes coated with 20 μg/ml Fibronectin (Sigma). For wound healing experiments, 80,000 cells were plated on dishes coated with 20 μg/ml Fibronectin within inserts (#80209, Ibidi) one day before the experiment. Inserts were removed before acquisitions. After seeding cells for 24 h, videomicroscopy was performed on an inverted Axio Observer microscope (Zeiss) equipped with a Pecon Zeiss incubator XL multi S1 RED LS (Heating Unit XL S, Temperature module, CO2 module, Heating Insert PS and CO2 cover), a definite focus module and a Hamamatsu camera C10600 Orca-R2. Images were acquired every 10 min for 24 h with a ×10 objective. Individual cells were tracked using ImageJ software with the Manual Tracking plug-in. DiPer software was used to analyse the single cell migration parameters[53]. To derive heat maps of speed and order parameter in wound healing, fields focused on only one edge of the wound were acquired, and average leading-edge positions perpendicular to the axis of the wound gave leading edge progression. The field of displacement vectors was obtained with the PIVlab software package[54] for Matlab (MathWorks). The window size was set to 32 pixels, i.e. 23.75 μm with a 0.75 overlap between windows. A time sliding window averaging fields over 4 frames (40 minutes) was used. Spurious vectors were filtered out on amplitude and were replaced by interpolated velocity from neighbouring vectors. The local order parameter was calculated using a previously published code[55].

## Statistical analysis

For statistical analysis, GraphPad Prism and Microsoft Excel were used. The Shapiro−Wilk normality test was performed. Two-tailed unpaired t-test was used for parametric data and Mann−Whitney test was used for non-parametric data. ANOVA followed by post hoc Tukey's multiple comparison test was used for parametric data and Kruskal−Wallis test followed by post hoc Dunn's multiple comparison test was used for non-parametric data.

Migration persistence for individual cells is evaluated based on the exponential decay and plateau fit according to equation ($y = (1 - b) * e^{-\frac{t}{a}} + b$), where $y$ is the migration persistence, $b$ is plateau value, $t$ is the time interval and $a$ is the decay constant. Then the related statistical analysis was conducted through custom-made R programmes, as previously described[41].

No statistical method was used to predetermine sample size. No data were excluded from the analyses. The experiments were not randomised. The Investigators were not blinded to allocation during experiments and outcome assessment.

### Reporting summary

Further information on research design is available in the Nature Portfolio Reporting Summary linked to this article.

## Data availability

Raw files of the LC-MSMS analyses and database searches have been deposited in PRIDE with the accession number PXD053609. Structural models can be accessed on modelarchive.org: NHSL3_i3S-14-3-3θ [https://www.modelarchive.org/doi/10.5452/ma-pptik], NHSL3_i2-Abi1_site1 [https://www.modelarchive.org/doi/10.5452/ma-pzi05], NHSL3_i2-Abi1_site2 [https://www.modelarchive.org/doi/10.5452/ma-e0o0t], NHSL3_i2-IRSp53 [https://www.modelarchive.org/doi/10.5452/ma-3kkf3], NHSL3_i2-MENA [https://www.modelarchive.org/doi/10.5452/ma-dfoes], and NHSL3_i2-VASP [https://www.modelarchive.org/doi/10.5452/ma-xrf33]. Source data are provided with this paper.

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

## Acknowledgements

This work was supported by grants from Agence Nationale de la Recherche (ANR-20-CE13-0016 and ANR-22-CE13-0041 to AMG, ANR-21-CE44-0009 to RG, ANR-24-CE44-4957 to AMG and RG), Fondation ARC pour la Recherche sur le Cancer (ARC PJA 2021 060003815 to AMG). Mass spectrometry equipment was subsidised by Conseil Régional d'Ile-de-France (Sesame N°10022268). This work was granted access to the HPC resources of IDRIS under the allocation 2024-AD010314343R1 to RG made by GENCI and to the BIOI2 platform resources at the I2BC.

## Author contributions

N.M.N. performed most experiments and analysed them. J.G. and D.J.Z. performed AlphaFold2 analyses of partners. A.I.F. and N.M.N. jointly performed collective migration experiments and cell imaging. N.R.

obtained KO cell lines and cloned NHSL3 isoforms. G.C. performed mass spectrometry. K.D.R. performed qRT-PCR analyses of NHSL3 isoforms. A.P., J.V., R.G., and A.M.G. supervised the work of their research groups. A.M.G. coordinated the work of the different groups. N.M.N. and A.M.G. jointly wrote the manuscript.

## Competing interests

The authors declare no competing interests.
