## [Transparent Peer Review file · Nature Communications]

NHSL3 Controls Single and Collective Cell Migration Through Two Distinct Mechanisms

Corresponding Author: Professor Alexis Gautreau

Version 0:

Reviewer comments:

Reviewer #1

(Remarks to the Author)

The authors have satisfactorily address all my comments. As such, I find their revised manuscript acceptable for publication.

Reviewer #2

(Remarks to the Author)

I am satisfied with the revisions that were made to this manuscript.

Reviewer #3

(Remarks to the Author)

The process of mass spectrometry experiments should be described with more details in the method section. The authors mention that 80% of the beads were subjected to MS analysis; however, it is unclear how did they process the beads.

What type of MS instrument and instrumental parameters were used for the analysis? And how was the data processed and analyzed?

What negative control was used for the comparison, and how did the authors filter out the background interfering proteins? Many of the identified proteins listed in table S2 are abundant cellular proteins, such as ribosomal proteins. I am concerned whether the identified proteins are genuine binders. In addition, the authors may highlight the actin regulators in table S2.

Version 1:

Reviewer comments:

Reviewer #3

(Remarks to the Author)

The authors have addressed all my comments.

REVIEWER COMMENTS

Reviewer #1 (Remarks to the Author):

The authors have satisfactorily addressed all my comments. As such, I find their revised manuscript acceptable for publication.

Reviewer #2 (Remarks to the Author):

I am satisfied with the revisions that were made to this manuscript.

Reviewer #3 (Remarks to the Author):

All the requested information concerning mass spectrometry was already present in the manuscript. It was, however, in the Supplementary Information that the reviewer must have overlooked. To alleviate this problem, we have moved all Material and Methods paragraphs into the main text. Below I just cut and pasted the corresponding information that answers the reviewer's inquiry.

The process of mass spectrometry experiments should be described with more details in the method section. The authors mention that 80% of the beads were subjected to MS analysis; however, it is unclear how they processed the beads.

The resins containing the immunoprecipitated samples were loaded onto a 10 kDa cutoff centrifugal filters (Microcon, Millipore-Merck) and washed with 500 μ L of buffer (50 mM, pH 8.0, AMBIC). Disulfide reduction was performed by adding to the centrifugal filters 200 μ L of a solution containing 10 mM dithiothreitol in AMBIC for 2 h at 37°C. Thiol alkylation was performed by adding to the previous samples 20 μ L of a solution containing 500 mM iodoacetamide in AMBIC for 30 minutes at room temperature. Reagents were removed by filtration and sample washed three times with 500 μ L of AMBIC. Proteins were digested with 0.5 μ g of trypsin/Lys-C (Promega) in 100 μ L of AMBIC overnight at 37°C. The resulting peptide mixture was filtered and acidified with trifluoro acetic acid at a final concentration of 0.1%. Technical triplicates were systematically analysed.

For each fraction, 6 μ L of sample was concentrated on a C18 cartridge (Dionex Acclaim PepMap100, 5 μ m, 300 μ m i.d. x 5 mm) and eluted on a capillary reverse-phase column (nanoEaze M/Z Peptide CSH C18, 1.8 μ m, 75 μ m i.d. x 25 cm) at 220 nL/min, with a gradient of 2% to 38% of buffer B in 60 min (buffer A: 0.1% aq. Formic Acid/Acetonitrile 98:2 (v/v); buffer B: 0.1% aq. Formic Acid/Acetonitrile 10:90 (v/v)).

What type of MS instrument and instrumental parameters were used for the analysis?

Liquid chromatography was coupled with a quadrupole-Orbitrap mass spectrometer (Q Exactive HF, ThermoFisher Scientific) using a Top 20 data-dependent acquisition MS experiment: 1 survey MS scan (400-2,000 m/z; resolution 70,000) followed by 20 MS/MS scans on the 20 most intense precursors (dynamic exclusion of 30 s, resolution 17,500).

And how was the data processed and analyzed?

Protein identification was performed with MaxQuant search engine v.1.5.3.30 against the human Swiss-Prot database (updated in 05/2022), with the following parameters: methionine oxidation, asparagine/glutamine deamidation as variable modifications, cysteine carbamidomethylation as fixed modification, first search error tolerance 20 ppm, main error tolerance 6 ppm, MS/MS error tolerance 20 ppm, FDR 1%. Quantification was performed in label-free LFQ normalization mode using at least 2 razor or unique peptide per protein. Quantities were estimated using LFQ intensities.

What negative control was used for the comparison, and how did the authors filter out the background interfering proteins? Many of the identified proteins listed in table S2 are abundant cellular proteins, such as ribosomal proteins. I am concerned whether the identified proteins are genuine binders.

Proteins detected in at least 2 of 3 replicates, and not detected in any of the control samples, Flag-GFP alone, were considered specific NHSL3 partners.

NHSL3, like the WAVE complex, indeed interacts with many elements of the translation machinery. In the case of the WAVE complex, a coupling between the generation of membrane protrusions and mRNA translation has been demonstrated (Mardakheh ... Marshall, Dev Cell 2015). For example, a subunit of the WAVE complex CYFIP1 interacts with mRNA binding proteins and translation initiation factors (Napoli... Bagni, Cell 2008; De Ruv-beis ... Bagni, Neuron 2013). So we suspect that even ribosomal proteins might be specific binders of NHSL3.

In addition, the authors may highlight the actin regulators in table S2.

We have now highlighted in yellow in Table S2 the known actin regulators, WAVE complex subunits, MENA/VASP and IRSp53, in both sheets corresponding to primary data of TAP with individual isoforms and with the knock-in.